# A Review of Sustainable Total Productive Maintenance (STPM)

Anouar Hallioui [1,*], Brahim Herrou [1,2], Polinpapilinho F. Katina [3], Ricardo S. Santos [4], Ona Egbue [3], Małgorzata Jasiulewicz-Kaczmarek [5,*], Jose Miguel Soares [6] and Pedro Carmona Marques [7,8]

1 Industrial Techniques Laboratory, Center for Doctoral Studies in Sciences and Techniques and Medical Sciences, Faculty of Sciences and Techniques of Fez, Sidi Mohamed Ben Abdellah University, Fez 30000, Morocco; brahimherrou@yahoo.fr
2 Superior School of Technology of Fez, Sidi Mohamed Ben Abdellah University, Fez 30000, Morocco
3 Department of Informatics and Engineering Systems, University of South Carolina Upstate, Spartanburg, SC 29303, USA; pkatina@uscupstate.edu (P.F.K.); egbueo@uscupstate.edu (O.E.)
4 Research Unit on Governance, Competitiveness and Public Policies, Santiago University Campus, University of Aveiro, 3810-193 Aveiro, Portugal; ricardosimoessantos84@ua.pt
5 Faculty of Management Engineering, Poznan University of Technology, 60-965 Poznan, Poland
6 ADVANCE-Research Center in Management, Instituto Superior de Economia e Gestão (ISEG), Universidade de Lisboa, 1349-017 Lisbon, Portugal; josesoares@iseg.ulisboa.pt
7 Instituto Superior de Engenharia de Lisboa (ISEL), Instituto Politécnico de Lisboa, 1959-007 Lisbon, Portugal; p4803@ulusofona.pt
8 Industrial Engineering and Management, Faculty of Engineering, Lusófona University, 1749-024 Lisbon, Portugal
* Correspondence: anouar.hallioui@gmail.com (A.H.); malgorzata.jasiulewicz-kaczmarek@put.poznan.pl (M.J.-K.)

**Abstract:** Sustainable Total Productive Maintenance (STPM) arose in 2021 as a promising, new concept to fill the lack of sustainability in Total Productive Maintenance (TPM) and allow companies to overcome its implementation barriers. It revolves around increasing the understanding of the systems approach and contributing to setting contemporary companies' sustainable ideology by supporting orientation toward sustainability from a sustainable maintenance perspective. However, STPM is still in its infancy and is viewed as a complement to the traditional TPM approach and is based on its pillars. Moreover, there is still a dearth of literature discussing STPM. This study aims to present STPM as a novel substitute for TPM while building its unique mechanism based on re-engineered fourth generation management (R4thGM). To pursue such a goal, 94 papers from Scopus, Web of Science, and Science Direct databases published in 2008–2023 were reviewed. This study's novelty comes from presenting STPM as the best-suited lean manufacturing and sustainability strategy for enhancing sustainable maintenance, encouraging contemporary maintenance (i.e., Industry 4.0 technology-based sustainable maintenance), and supporting second-era contemporary companies' orientation toward sustainability. Furthermore, based on recent studies, propositions are formulated to achieve STPM. Finally, research implications and future directions are presented.

**Keywords:** sustainable total productive maintenance; sustainability; sustainable maintenance; re-engineered fourth generation management; contemporary maintenance; second-era contemporary companies

## 1. Introduction

The involvement of companies in the UN 2030 Agenda for Sustainable Development, announced in 2015, creates an excellent opportunity for enterprises to strengthen the sustainability of their operations by integrating more robustly into the economic, environmental, and social situations where they operate [1]. Launching a new business generation oriented toward sustainability and customers (i.e., more contemporary organization generation) by creating re-engineered fourth generation management (R4thGM) in 2022 [2] was a turning point. This had significant implications for fostering the change of business

functions into contemporary functions (e.g., contemporary maintenance) and re-thinking lean manufacturing, production, and maintenance philosophies such as Total Productive Maintenance (TPM) to support organizations' sustainability in the second era of contemporary businesses characterized by Industry 4.0 technologies and increased consciousness of the circular economy. This new era began with the fourth industrial revolution in 2011. Figure 1 below shows a striking increase in research on Industry 4.0 and the circular economy in engineering from 2016 onward.

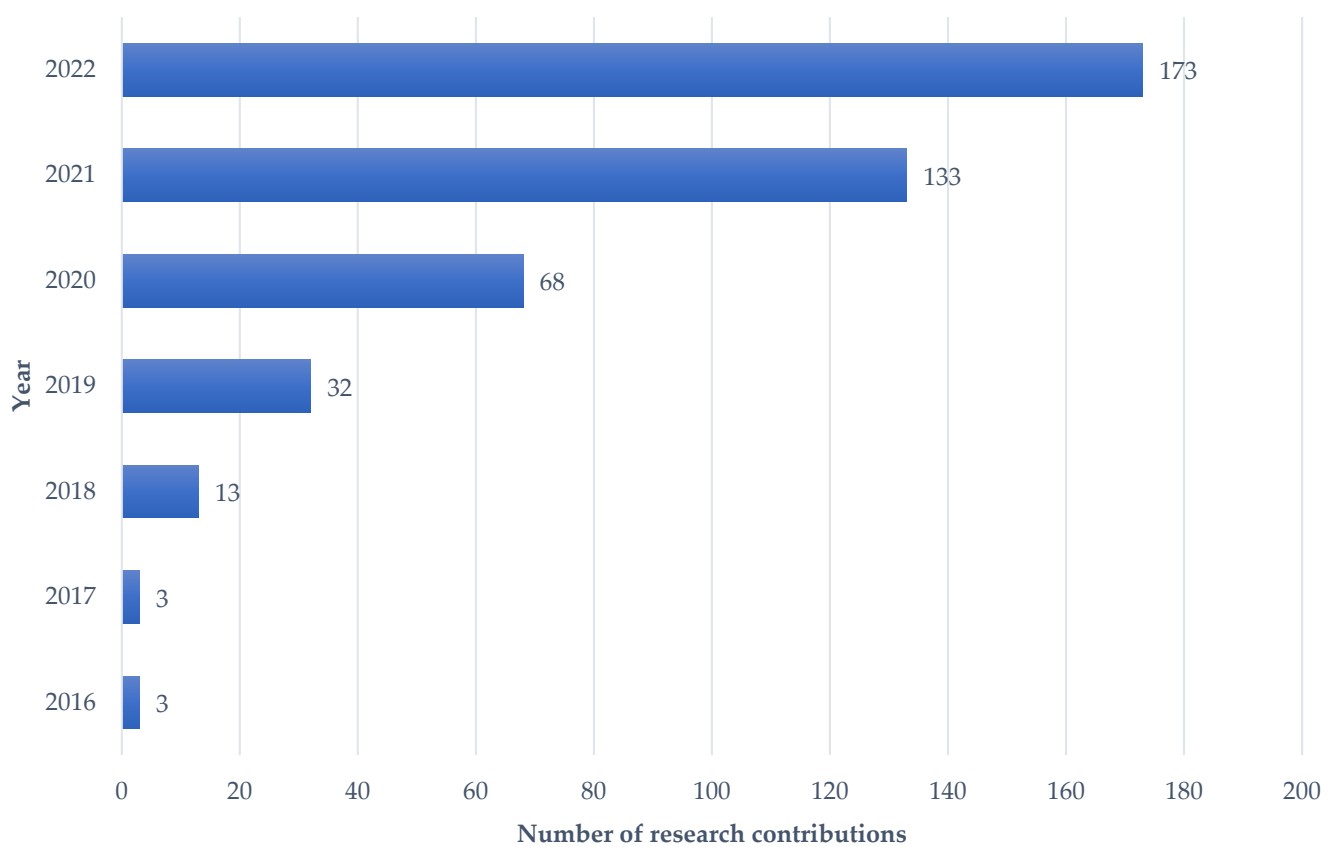

**Figure 1.** Research contributions (i.e., journal articles and conference papers) regarding Industry 4.0 and the circular economy in the field of engineering from 2016 onward according to Scopus-Elsevier.

Maintenance as a business function is an essential part of pursuing sustainable development and attaining the status of a sustainable company [3,4]. Therefore, from a systemic viewpoint, contemporary maintenance is an active and immersed part of contemporary business. The literature has recently shown the difference between contemporary organizations of the first and second eras. According to [2], contemporary businesses of the first era, which emerged in the 1990s, are customer-oriented. Joiner [5] introduced the concept of fourth generation management to enable these organizations to be managed as an open system to stakeholders. In comparison, contemporary businesses of the second era are more open and sustainable due to systems focused on sustainability and customers through the revolutionary management thinking enabled by R4thGM as a novel management paradigm. The first and most exhaustive definition of contemporary maintenance in the literature was provided by the author of [6] as a concept that goes beyond the set of operations, focusing on addressing malfunctions and breakdowns and equipment preservation. The author points out that modern maintenance is about long-term strategic planning that covers all stages of the product life cycle, takes into account and foresees changes in economic, environmental, and social trends, and benefits from cutting-edge technologies. Thus, contemporary maintenance can be defined as Industry 4.0-based sustainable maintenance.

Sustainable maintenance is a new challenge for companies realizing concepts of sustainable development. These concepts include proactive maintenance operations aimed at reaching balance in economic (losses, consequences, benefits, etc.), environmental (ReSOLVE framework or the circular economy dimensions, 6R or Reduce, Reuse, Remanufacture, Recycle, Recover, and Redesign, etc.), and social dimensions of sustainability (welfare, maintenance employees' satisfaction, etc.) [6]. Sustainable maintenance is achieved by considering the connection between maintenance, sustainability, and Industry 4.0, as well as the critical role of digitalization technologies [7]. Thus, maintenance strategies ensuring equipment availability, reliability, and safety, such as TPM, are crucial in sustainable manufacturing. Furthermore, many researchers [8–15] have recently suggested combining lean manufacturing tools and TPM with technologies from Industry 4.0 to create a lean 4.0 landscape and lead manufacturing companies toward sustainability.

Due to recent debate regarding the combination of lean–green and sustainability tools starting in 2015 (Figure 2) and the transition from TPM toward STPM (i.e., Sustainable Total Productive Maintenance), TPM was scaled up to Green TPM in 2020 to deal with sustainability and maintenance simultaneously, therefore enhancing manufacturing and environmental performance for manufacturing companies by covering green aspects, e.g., green training, green maintenance, etc. [16]. These changes can lead to green manufacturing emphasizing concerns related to environmental pollution, including wastewater management and supply, environmental protection, pollution control, regulatory compliance, waste recycling, and others [17].

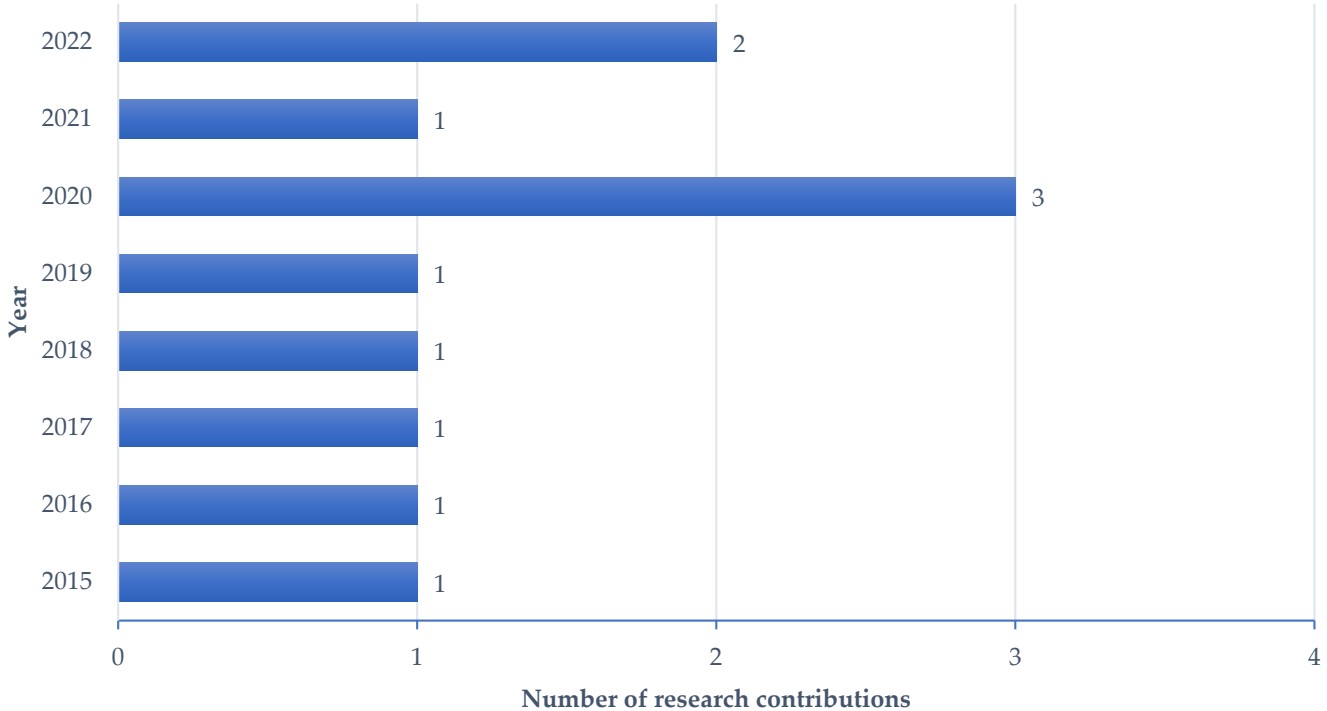

**Figure 2.** Research contributions (i.e., journal articles and conference papers) on sustainability and lean–green manufacturing tools from 2015 onward according to Scopus-Elsevier.

STPM arose in 2021 as a promising new concept to fill TPM's lack of sustainability and overcome its implementation barriers. STPM revolves around increasing the understanding of the systems approach (i.e., all working within a system) and contributing to setting contemporary companies' sustainable ideology by supporting their orientation toward sustainability from a sustainable maintenance perspective [18]. Still, considering the capability of TPM practices in making maintenance operations more sustainable, STPM is tackled as an innovative methodology that complements TPM practices by integrating lean and sustainable manufacturing principles to improve manufacturing organizations'

sustainability [19]. The motivation for our study, which builds on previous research by the authors of [18], is to demonstrate that STPM is best suited for supporting second-era contemporary manufacturing businesses' orientation toward sustainability by enhancing sustainable maintenance.

STPM is still misunderstood among academics and practitioners and is considered a complementary technique to Japanese or classical TPM. Furthermore, there remains a scarcity of literature discussing STPM. Therefore, our study's core challenge is to focus on the literature to make STPM a compromise between sustainable maintenance, contemporary maintenance, R4thGM, and second-era contemporary manufacturing businesses. Consequently, this work aims to present STPM as a substitute for TPM while building its unique mechanism based on R4thGM. The study focuses on supporting manufacturing companies' sustainability orientation from a sustainable maintenance viewpoint in the second era of contemporary organizations. The research methodology is described in Section 2. Section 3 discusses the chronological context of the creation of STPM and its suggested unique architecture. Section 4 presents propositions to achieve STPM and research implications. Finally, conclusions and future research directions are given in Section 5.

## 2. Materials and Methods

This study utilizes a three-stage research methodology (Figure 3). The first stage involves a framework proposal and a step related to pioneering the creation and presentation of the innovative ideas of STPM. It was the initial step toward thinking outside TPM's box and designing STPM's mechanism as a substitute (Section 3.4). The first stage also included receiving feedback from other researchers at an international conference. The second stage resulted in the creation of R4thGM and its tool, Hallioui's triangle (Figure 4), to fill the void created by the absence of a management style in the literature to help businesses become oriented toward sustainability and customers. Finally, the third stage shows the process of the literature review to address the following research gaps: (i) in the literature, STPM is still misunderstood—it is viewed as a complement to Japanese or classical TPM while being based on its eight pillars; (ii) there is a scarcity of literature discussing STPM (Table 1).

A literature review is ideal for synthesizing study outcomes and developing theoretical frameworks [2,20]. This study uses Scopus, Web of Science, and Science Direct databases to identify peer-reviewed papers as crucial secondary data sources. Six categories consisting of a total of 32 search terms were considered (Figure 3 and Table 1). The inclusion criteria for the study required that papers (i) are indexed in Scopus or Web of Science; (ii) are non-redundant journal and conference papers published between 2008 and 28 February 2023; (iii) contain at least one of the highlighted concepts in the title, abstract, or keywords (STPM, TPM, contemporary maintenance, re-engineered fourth generation management, or sustainable maintenance); and (iv) relate to the concepts of STPM, TPM, or contemporary maintenance and discuss them in the context of manufacturing or sustainable businesses (Figure 3). RStudio [21] merged bibliographic data frames from Scopus and Web of Science and removed duplicate documents for each search term (Table 1). Based on the inclusion criteria, 94 papers were selected from 684,611 documents published in 2008–2023. Of the selected papers, 34% are conference papers while 66% are journal articles, and 77% were published between 2018 and 2023. The journals with the most articles published during 2018–2023 are the Journal of Quality in Maintenance Engineering, with four articles, followed by Sustainability (Switzerland) and the TQM Journal, with three articles each.

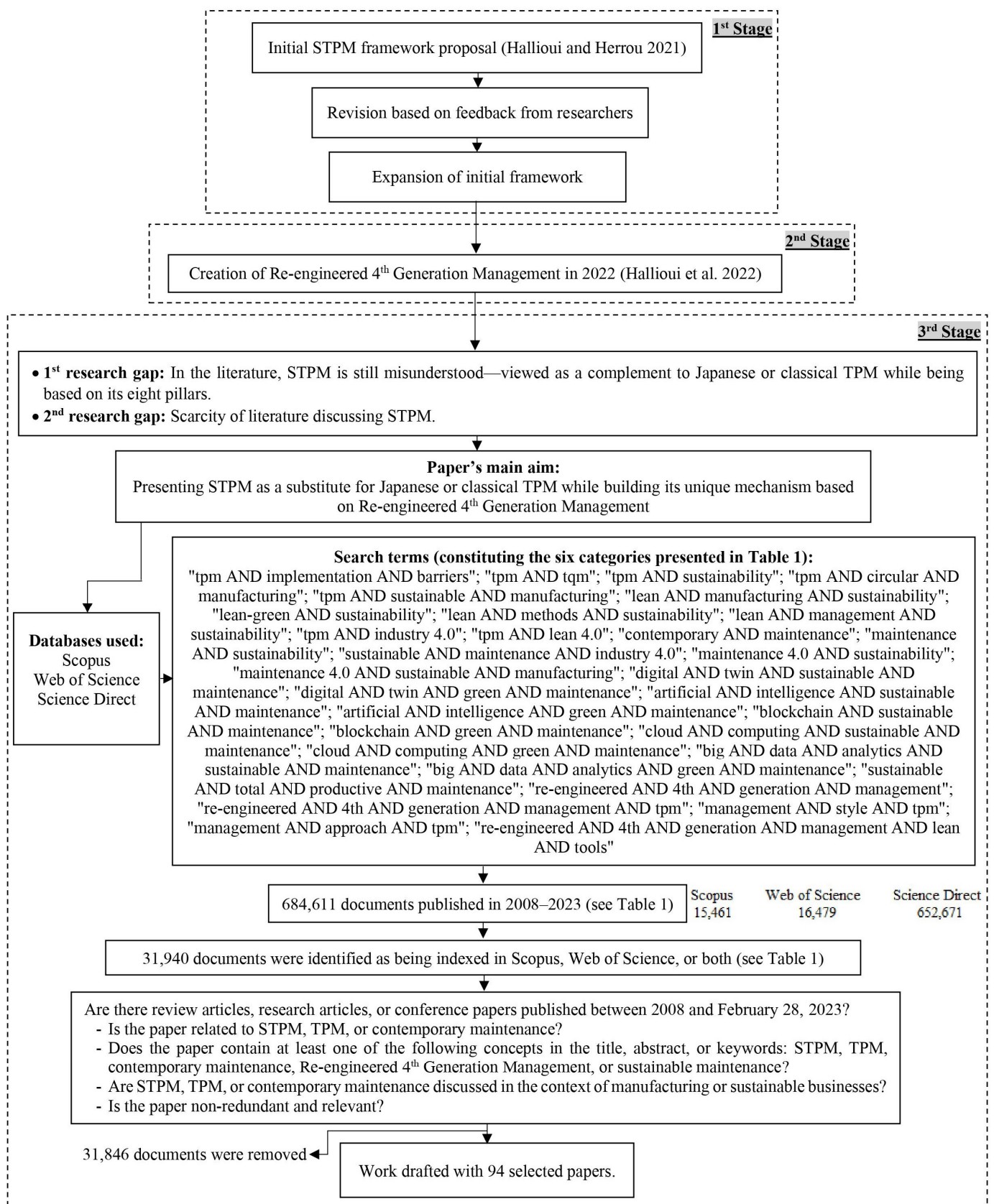

**Figure 3.** Diagram of research methodology. The authors of [2,18] tackled stages 1 and 2.

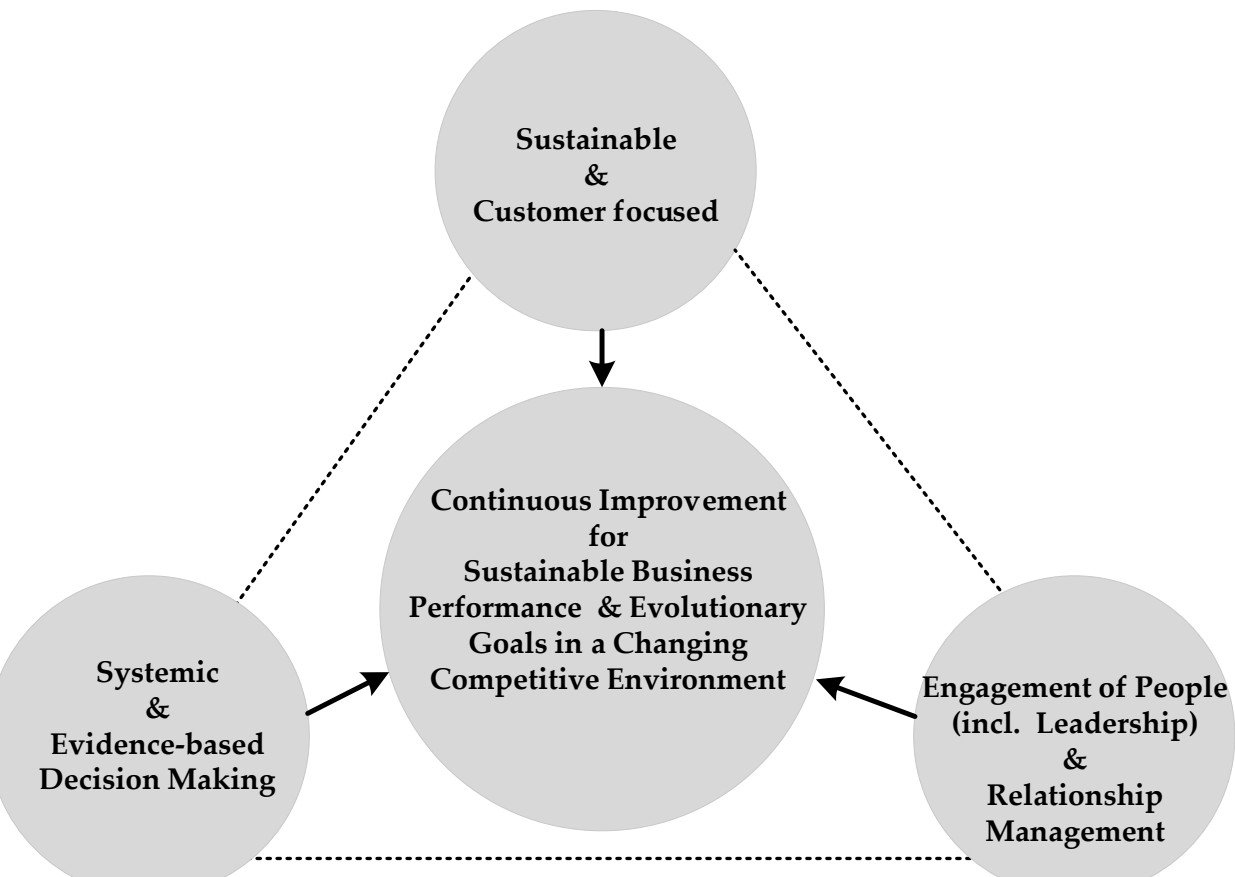

**Figure 4.** Hallioui's triangle—the basis for R4thGM (based on [2]).

**Table 1.** The number of relevant papers per category of search terms used in Scopus, Web of Science, and Science Direct.

| Categories of Search Terms | Search Terms | Scopus (Publication Year 2008–2023) | Web of Sciences (Publication Date from 1 January 2008 to 28 February 2023) | Science Direct (Publication Year 2008–2023) | Total Documents Identified | Duplicates in Scopus and Web of Science | Number of Relevant Papers | Relevant Papers |
|---|---|---|---|---|---|---|---|---|
| 1st category | tpm AND implementation AND barriers | 38 | 35 | 1113 | 1652 | 22 | 60 | [22–81] |
| | tpm AND tqm | 106 | 76 | 284 | | 43 | | |
| 2nd category | tpm AND sustainability | 50 | 32 | 3406 | 102,355 | 25 | 12 | [14,16,17,82–90] |
| | tpm AND circular AND manufacturing | 3 | 3 | 543 | | 2 | | |
| | tpm AND sustainable AND manufacturing | 31 | 17 | 1171 | | 12 | | |
| | lean AND manufacturing AND sustainability | 816 | 706 | 14,144 | | 414 | | |
| | lean–green AND sustainability | 90 | 74 | 14,624 | | 49 | | |
| | lean AND methods AND sustainability | 585 | 505 | 38,384 | | 291 | | |
| | lean AND management AND sustainability | 952 | 1189 | 25,030 | | 466 | | |

**Table 1.** *Cont.*

| Categories of Search Terms | Search Terms | Scopus (Publication Year 2008–2023) | Web of Sciences (Publication Date from 1 January 2008 to 28 February 2023) | Science Direct (Publication Year 2008–2023) | Total Documents Identified | Duplicates in Scopus and Web of Science | Number of Relevant Papers | Relevant Papers |
|---|---|---|---|---|---|---|---|---|
| 3rd category | tpm AND industry 4.0 | 40 | 32 | 703 | 1075 | 19 | 12 | [8–13,15,91–95] |
|  | tpm AND lean 4.0 | 15 | 15 | 270 |  | 6 |  |  |
| 4th category | contemporary AND maintenance | 3046 | 3096 | 46,766 | 538,402 | 1790 | 7 | [4,7,96–100] |
|  | maintenance AND sustainability | 8227 | 9251 | 314,900 |  | 4949 |  |  |
|  | sustainable AND maintenance AND industry 4.0 | 143 | 151 | 18,127 |  | 57 |  |  |
|  | maintenance 4.0 AND sustainability | 130 | 149 | 39,746 |  | 64 |  |  |
|  | maintenance 4.0 AND sustainable AND manufacturing | 79 | 103 | 13,140 |  | 38 |  |  |
|  | digital AND twin AND sustainable AND maintenance | 80 | 57 | 3785 |  | 25 |  |  |
|  | digital AND twin AND green AND maintenance | 18 | 15 | 2853 |  | 4 |  |  |
|  | artificial AND intelligence AND sustainable AND maintenance | 216 | 210 | 12,922 |  | 72 |  |  |
|  | artificial AND intelligence AND green AND maintenance | 84 | 74 | 8897 |  | 24 |  |  |
|  | blockchain AND sustainable AND maintenance | 44 | 34 | 2696 |  | 18 |  |  |
|  | blockchain AND green AND maintenance | 16 | 12 | 1595 |  | 6 |  |  |
|  | cloud AND computing AND sustainable AND maintenance | 93 | 100 | 10,537 |  | 40 |  |  |
|  | cloud AND computing AND green AND maintenance | 158 | 79 | 8160 |  | 33 |  |  |
|  | big AND data AND analytics AND sustainable AND maintenance | 43 | 50 | 16,896 |  | 20 |  |  |
|  | big AND data AND analytics AND green AND maintenance | 22 | 11 | 11,607 |  | 7 |  |  |
| 5th category | sustainable AND total AND productive AND maintenance | 65 | 65 | 27,369 | 27,499 | 31 | 2 | [18,19] |

**Table 1.** *Cont.*

| Categories of Search Terms | Search Terms | Scopus (Publication Year 2008–2023) | Web of Sciences (Publication Date from 1 January 2008 to 28 February 2023) | Science Direct (Publication Year 2008–2023) | Total Documents Identified | Duplicates in Scopus and Web of Science | Number of Relevant Papers | Relevant Papers |
|---|---|---|---|---|---|---|---|---|
| 6th category | re-engineered AND 4th AND generation AND management | 1 | 1 | 8471 | | 1 | | |
| | re-engineered AND 4th AND generation AND management AND tpm | 0 | 0 | 77 | | 0 | | |
| | management AND style AND tpm | 8 | 17 | 312 | 13,628 | 4 | 1 | [2] |
| | management AND approach AND tpm | 278 | 320 | 3325 | | 140 | | |
| | re-engineered AND 4th AND generation AND management AND lean AND tools | 0 | 0 | 818 | | 0 | | |
| | The total number of duplicates and non-relevant documents was 684,517 | | | | | | | |

## 3. STPM as a Substitute for Japanese or Classical TPM

*3.1. Chronological Context of the Creation of STPM*

3.1.1. Emergence of TPM

Companies began to worry about offering higher quality products from World War II and the last decades of the mass production era (i.e., the second industrial revolution or pre-lean automation era) (Figure 5). Japanese industries were the pioneers in improving quality based on the ideas from the United States [101]. They transformed these ideas into successful practices to make products manufactured in Japan known for their superior quality and exported them to Western industrial countries in large quantities [102]. Japanese fathers of manufacturing and management systems (e.g., [102,103]) never denied their learning from the US manufacturing empire to import ideas and put them into practice in Japan, nor their struggles, successes, frustrations, and surprises to lead the transformation of Japanese industry. Moreover, the world wars during the 20th century played a centric role in creating energy transitions given the growth of oil as a significant energy source, which accelerated after the Second World War in North America and Europe [104]. Therefore, the improvement in the quality of products and the energy transition were the significant factors contributing to increased competitiveness among companies from the Second World War onwards. During the 1950s, the author of [105] introduced the systems approach to address the complexity of the interacting elements of systems.

During the second half of the 20th century, researchers (e.g., [106–108]) studied the systems approach and its tools, such as systems theory (i.e., general system theory), system analysis, and cybernetics [109]. These studies led to a better understanding among companies of the complexities of working within a system. It is crucial to indicate that the author of [110] and then the authors of [111] were the pioneers in emphasizing the concept of the systems approach to understand the complex meaning of classical TPM.

While considering quality control as a classical quality management approach starting in the United States in the 1920s [112], which was based on statistical and mathematical techniques [113], the following factors should be taken into account: (i) the introduction of quality control methods (e.g., control charts, sampling inspection techniques, etc.) in Japanese factories in the early 1950s [112]; (ii) the appearance of the Japanese style of quality control as a thought revolution in management [114], which is termed Total Quality Control (TQC), though after Feigenbaum's American TQC [115,116]; (iii) the development of a discipline under the name Reliability Theory from World War II [117] to analyze the failure of complex, multi-component engineered systems, which typically defines exceeded safety thresholds [118,119]; and (iv) the advent of preventive maintenance and

productive maintenance during the 1950s and 1960s, respectively [102], all of which had crucial roles in creating TPM at the Japan Institute of Plant Maintenance in the 1970s, as an approach denoted to develop the preventive maintenance methodologies created by the Americans [110]. TPM's creation came after the author of [102] studied American preventive maintenance in 1950 and after his first visit to the United States in 1962; every year after then, he visited Western (i.e., American and European) manufacturers to study their manufacturing facilities and learn more about preventive maintenance systems. Based on his observations, the author of [102] developed TPM and introduced it in Japan in 1971. Those were all historical processes, leading to the reason for one of the first and most famous definitions of TPM, which was given by the author of [102]: like TQC, which is company-wide total quality control, TPM is equipment maintenance performed on a company-wide basis. He considered TPM the fourth stage after the development of the first three stages, which include breakdown maintenance, preventive maintenance, and productive maintenance. In breakdown maintenance, an old maintenance strategy that characterized the period before the 1950s, the repair is performed after equipment stoppage, failure, or severe performance decline [79,120]. Preventive maintenance was introduced in 1951 to systematically intervene with equipment before anomalies to maintain and improve its working conditions and operational performance. Lastly, productive maintenance is the best economic maintenance that enhances the company's productivity by lowering the overall cost of the equipment during the span of its life (i.e., during the stages of design, manufacture, operation, and maintenance) while also reducing the losses brought on by its degradation [79].

### 3.1.2. After the Appearance of TPM and the Advent of STPM

According to [112], a remarkable increase in concerns for safety and reliability in Japanese TQC characterized the 1960s before TPM emerged. Japanese manufacturers regarded quality assurance based on reliability as one of the foundations of quality control education in the early 1970s. In the same decade, during the first oil crisis in the fall of 1973, the Toyota production system supported by the just-in-time system and autonomation (i.e., automation with a human touch) began attracting the Japanese industry's attention after its design and implementation soon after World War II [103]. The Toyota production system and, ultimately, lean production was the Japanese substitute that led the Toyota Motor Company out of the American mass production model that was unworkable in Japan. Eiji Toyoda and his production genius Taiichi Ohno's viewpoint believed in Japanese manufacturing's values, culture, and unicity in the context of their company and the circumstances of the country's crisis during the post-war era [121]. Just-in-time production, an innovation pioneered at Toyota in the 1950s, was first embraced by Western firms in the early 1980s [122]. In just-in-time production, only the necessary products at the required times in the essential quantities are produced while maintaining the stock on hand at a minimum [123]. Mr. Taiichi Ohno, a Toyota executive and the creator of Toyota's just-in-time production system, also identified the first seven kinds of Muda—overproduction, waiting, transportation, processing itself, stock on hand, movement, and the creation of defective goods [122]. According to [122], the Japanese concept of Muda means waste, specifically any human activity that uses resources without creating value.

Independent organizations began developing standards in the last decades of the 20th century to support the rise of Quality Management Systems (QMS); at the time, the concept of Total Quality Management (TQM) originated as an American response to Japan's quality revolution [124]. In the literature, TQM is often discussed with or without TPM or JIT in a framework of lean manufacturing or continuous improvement to achieve world-class manufacturing [23,35,37,41,51–55,58–64,66–70,72,74,77,78,81,125], and the literature is inaccurate in encapsulating TQM and its country of origin (i.e., US or Japan) [115,126]. TQM is a company culture in which all employees actively participate in quality improvement [127]. This TQM definition enabled the authors of [128] to show that there do not seem to be any contradictions between TQM and lean production objectives,

which proves that the roots of TQM, born in the late 1980s, can be linked to the development of Japanese quality.

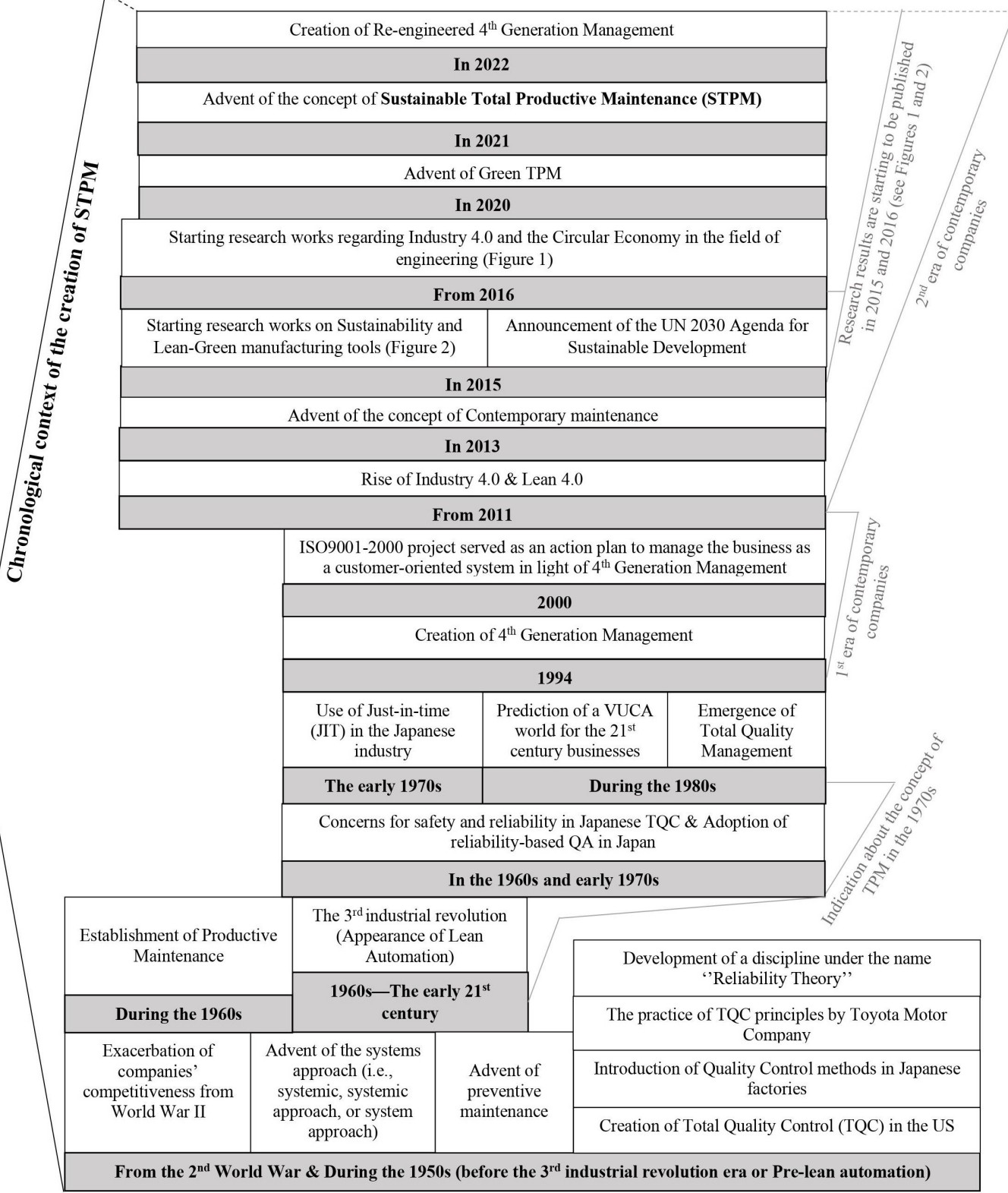

**Figure 5.** The chronological context of the creation of Sustainable Total Productive Maintenance.

The early anticipation of a volatile, uncertain, complex, and ambiguous world (VUCA world) for 21st-century businesses in the 1980s [129] led company leaders to rethink how they tackled culture, management, quality, and technology-related concerns. Indeed, it was a solid sign to shift their leadership paradigm in order to survive in a more turbulent future. It has been proposed that companies should rethink their strategies and processes to consider a highly competitive and dynamic landscape [2,15]. Furthermore, in the last decades of the 20th century, business managers and workers needed to learn to trust each other and live with and embrace a system-oriented, data-based customer focus [2]. Nevertheless, the traditional character stayed predominant in business management; in other words, companies were not capable of transitioning from traditional to contemporary organizations at that time, and only became capable after the creation of the so-called fourth generation management style in the 1990s [5]. According to the authors of [5]'s own words, the key elements of fourth generation management are a dedication to quality as defined by the customer (i.e., customer focus), a scientific approach to rapid learning, and the creation of team-spirited relationships (i.e., all one team). These three elements (quality, scientific approach, and all one team) are known as Joiner's triangle. It aims at filling the gap of the lack of a clear explanation of the basic principles that underlie and unify the seemingly diverse approaches existing before the 1990s, such as TQM, continuous improvement, re-engineering, time-based competition, and visionary leadership [5]. The ISO 9001-2000 project [130] was an action plan to enable companies to manage quality based on fourth generation management by suggesting the eight quality management principles, including the process approach applied in managing the organization as a customer-oriented system, customer focus, leadership, relationship management, continuous improvement, evidence-based decision making, and the engagement of people.

Looking back to [129]'s 1980s postulate that it is hard to think of any 21st-century industry or government agency that will not find itself heavily dependent on technology in its operations, products, or services, Industry 4.0 emerged in 2011 as a major catalyst for waste removal and productivity growth [131], which supports that early prediction. Industry 4.0 forges new industrial production by linking machines and people for a faster exchange of information and is supported by web technology and intelligent systems [94,132]. In other words, it looks to connect the physical and virtual worlds in industrial production and has acquired popularity, as well as several opportunities, strategies, and business models that can be strengthened using digitalization technology [133]. Industry 4.0 has gained significant attention in recent years due to its potential to transform manufacturing strategies substantially [12,134,135]. Besides Industry 4.0, the interaction between lean manufacturing and Industry 4.0 paradigms has created great debate among researchers. It can be materialized by a finalized two-way relationship, i.e., lean tools (TPM, TQM, JIT, etc.) enable Industry 4.0 implementation while Industry 4.0 advances lean manufacturing, and their combination results in the concept of Lean 4.0 [13,136]. Lean 4.0 is the fourth phase of lean manufacturing evolution and the successor of lean automation, which began with the third industrial revolution in the 1960s [137].

As detailed in Section 1 and shown in Figure 5, after the debut of Industry 4.0 in 2011, notably after (i) the advent of the concept of contemporary maintenance in 2013; (ii) the announcement of the UN 2030 Agenda for Sustainable Development in 2015; (iii) the striking increase in research works on "Sustainability and Lean–Green manufacturing tools" (Figure 2) and "Industry 4.0 and the circular economy in the field of engineering" (Figure 1) in 2015 and 2016, respectively; and (iv) the creation of the concept of Green TPM in 2020 by the authors of [16], STPM was created in 2021 by the authors of [18], who first presented it as an innovative concept at an international conference in Italy (Figure 3). In 2022, the authors of [2] created R4thGM as a new management style to enable organizations to be oriented toward sustainability and customers in the context of Industry 4.0, the circular economy, competitiveness, and diverse stakeholders, which can serve as a basis for STPM and a prerequisite for its achievement (Sections 3.4 and 4.1.1).

*3.2. Barriers to TPM*

Researchers have dedicated more than 50 years to determining the causes of resistance to change and how to deal with it because it can significantly impact whether improvements succeed or fail [138]. The attempts to change reality for an organization have never been easy to implement given the tough opposition (i.e., complex resistance) imposed by several resources and concerns. In line with the trends of the 21st century, it seems particularly difficult to introduce changes in enterprises from the manufacturing sector [92]. The authors of [139] stated that more than 70% of attempts to bring about change in organizations fail because of factors including a lack of focus on business processes, disregard for the values and beliefs of employees, organizational culture traits, ineffective attempts to promote change in specific locations, inadequate leadership, high employee resistance, inaccurate estimation of a temporary resource, etc. The case of the Toyota Motor Company before they created their production system is a good example (Section 3.1.2). That complex resistance is the set of constraints originating from the mindset of personnel and diverse stakeholders, equipment technology, the methods used, and financial resources constituting the external and internal environments of the business as a system (i.e., whole).

Numerous manufacturing companies attempted TPM in the past, but they failed because of an insufficient understanding of the obstacles to its effective implementation [28]. The literature has long provided many works on worldwide TPM implementation barriers; [22,28,42,49,57,65,71,73,75,76] are the most recent and accurate research works analyzing the barriers to TPM implementation across manufacturing companies. These works emphasized and detailed the following ten critical barriers to TPM, which were ranked in order of criticality in [75] as follows: (1) lack of top management commitment and support; (2) lack of training and education; (3) lack of motivation; (4) employee resistance; (5) cultural resistance; (6) failure to allow sufficient time for the evolution; (7) poor relationship between the production and maintenance department; (8) lack of communication; (9) financial constraints; and (10) lack of understanding and knowledge of TPM. The most significant success factor in implementing TPM is the involvement and commitment of top management [30,49,57,71,75,76].

The literature categorizes these barriers into numerous families, such as cultural, organizational, behavioral, technological, financial, departmental, and operational barriers [42,78]. However, this categorization or understanding is reductionist (i.e., pragmatic in that it comes from analytical thinking that reduces the organization into its functions or departments), uncertain, and ineffective in overcoming TPM's implementation barriers since it is confusing for companies. For instance, according to [42], a lack of training and education is regarded as an organizational, behavioral, cultural, and technological barrier. Still, employee resistance is considered a behavioral, cultural, operational, and organizational barrier.

All are managerial (i.e., organizational). On the one hand, management issues can generate behavioral, administrative, cultural, technological, departmental, operational, and financial problems, inhibiting company development and adaptation to the changing context of the business. On the other hand, through management, firms can eliminate all those categories of barriers. Figure 6 presents a fault tree of the failure to implement TPM as a philosophy across manufacturing businesses worldwide. It is a logic diagram that shows, through the recategorization of TPM's barriers, the root causes of problems related to the deployment of TPM by companies over time, from its creation until today. TPM is still viewed from an operational point of view. In other words, the management style and its ideological implications do not matter yet for organizations implementing TPM.

Barriers in the organization's context mean that walls exist between its functions. Thus, their existence directly reflects management issues, such as organizational reductionism (i.e., bureaucracy) or an inadequate management style for the implementation of improvement projects such as TPM. Besides the findings of [75] regarding barrier rankings, it is important to add that top management can effectively contribute to manufacturing performance improvements by providing an effective structure for TPM implementation [80].

Moreover, two support practices (top management and leadership and human resource development) benefitted from adopting technological techniques such as TPM [49,50]. In other words, a high level of top management maturity is necessary to implement TPM successfully [56,76]. Thus, relating lean manufacturing tools such as TPM to the company's management style (e.g., R4thGM) is necessary to implement them within the manufacturing business as a system in general and more contemporary business in particular (i.e., second-era contemporary business).

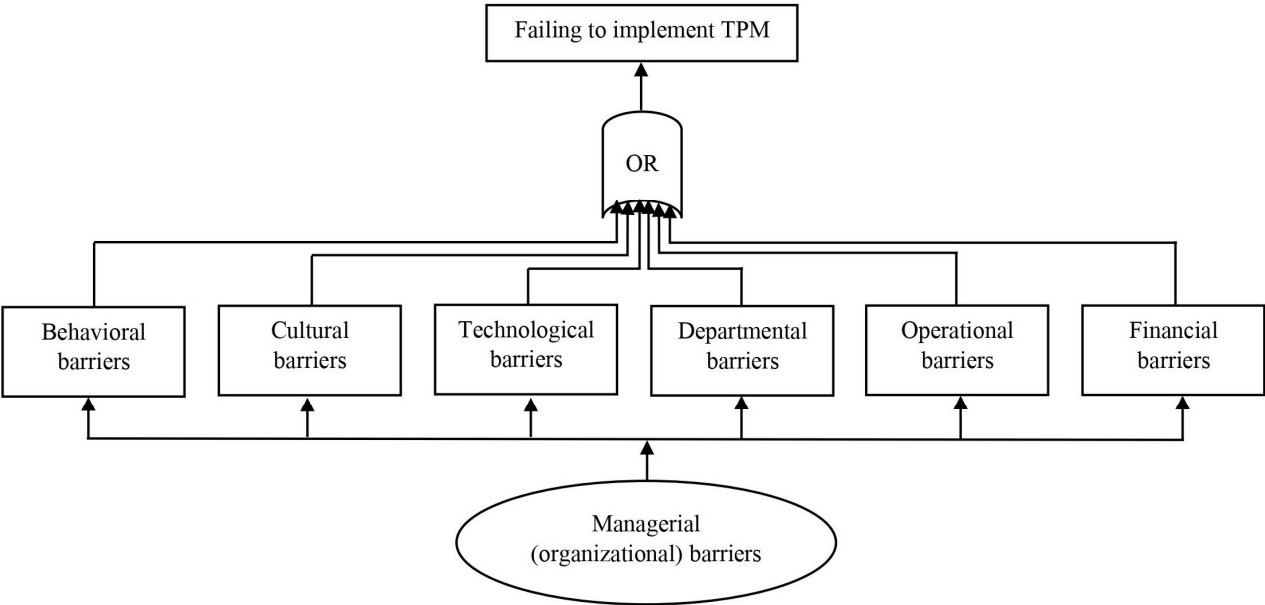

**Figure 6.** Fault tree of the failure to implement TPM within manufacturing companies.

### 3.3. Lack of Sustainability in the Concept of TPM

Businesses of the twenty-first century must prioritize sustainability [2]. The 2021 State of Green Business Report showed that, at last, sustainability has emerged from the shadows to become vital to corporate success. Many of the largest businesses in the world increasingly consider sustainability as essential to reducing risk, boosting resilience, strengthening competitiveness, and generating new opportunities. In terms of critical findings, in 2019, 90% of major US companies released a sustainability report, up from 86% in 2018 and 20% in 2011 [140]. Furthermore, sustainability is firmly ingrained in companies' objectives and strategies based on the 2019 State of Sustainable Business Report [141]. The transition toward more sustainable business practices generally necessitates changes to an organizations' products, services, processes, policies, and resources [142]. Today's businesses must include sustainability in their strategies to ensure long-term survival, growth, and profitability in a competitive and constantly changing world [143]. Since its creation in Japan in the 1970s, TPM has known many definitions and several perspectives (Table 2). However, academics failed to introduce sustainability into the concept of TPM to forge STPM until 2021.

**Table 2.** Definitions of TPM and STPM in the period of 2020–2022. All chosen papers have undergone peer review and are indexed in Scopus/Web of Science/Science Direct.

| Year | Definition of TPM/STPM | STPM/Classical TPM | Source |
|---|---|---|---|
| | TPM is a lean manufacturing practice that can improve inventory turnover performance. | Classical TPM | [26] |
| | TPM is a lean manufacturing tool that directly impacts social, economic, and environmental sustainability. | Classical TPM | [82] |
| | TPM is a lean manufacturing tool associated with machinery and equipment tools and directly and positively affects environmental sustainability. | Classical TPM | [83] |
| | TPM is an essential lean method without a significant relationship with firms' sustainable performance. | Classical TPM | [85] |
| | TPM is among the critical green lean six sigma practices that allow organizations to manage waste effectively, conserve resources, control air emissions, and improve environmental and workplace safety. | Classical TPM | [84] |
| | TPM is a methodology that allows businesses to improve their productivity by focusing on occupational ergonomics. | Classical TPM | [32] |
| | TPM is a methodology that comprises all maintenance policies, such as predictive, preventive, and corrective maintenance, to reduce subsystem failures, reduce system downtime, and improve reliability and productivity. | Classical TPM | [144] |
| | TPM is a strategy that can potentially increase business machinery efficiency by minimizing downtime, speed, and quality losses. | Classical TPM | [25] |
| | TPM is a strategy to manage equipment, reduce waste and lead time, and enhance competitiveness. | Classical TPM | [27] |
| 2022 | TPM is a reliable maintenance strategy that has been incorporated into the work culture of many large industries. | Classical TPM | [28] |
| | TPM is a comprehensive strategy, provided the organization's employees can participate in the operation and maintenance activities. | Classical TPM | [145] |
| | TPM is a philosophy that can be used as a long-term strategy to improve productivity in an organization. | Classical TPM | [30] |
| | TPM is a technique that can be incorporated into lean, smart manufacturing by combining it with Industry 4.0 technologies. | Classical TPM | [10] |
| | TPM is a problem-solving technique that contributes to improving manufacturing businesses through enhanced productivity and the cutting of costs. | Classical TPM | [29] |
| | TPM is a shop floor practice that can be digitalized through Industry 4.0 to lead a manufacturing firm toward sustainability. | Classical TPM | [8] |
| | TPM is a quality system that can improve the business's overall performance. | Classical TPM | [31] |
| | TPM is an economical maintenance variant that ensures stability, quality, and the maximization of production efficiency. | Classical TPM | [24] |
| | TPM is company-wide preventive maintenance. | Classical TPM | [91] |
| | TPM is a dynamic capability that forges a new bundle with Industry 4.0 and the circular economy to ensure sustainable performance for manufacturing businesses. | Classical TPM | [9] |
| | STPM is a complement to TPM practices. It could become an essential tool for more sustainable manufacturing. | STPM | [19] |

**Table 2.** *Cont.*

| Year | Definition of TPM/STPM | STPM/Classical TPM | Source |
|---|---|---|---|
| 2021 | TPM is a lean manufacturing approach that can be combined with green manufacturing approaches (e.g., 6R techniques—Reduce, Reuse, Remanufacture, Recycle, Recover, and Redesign) and Industry 4.0 technologies to achieve optimized and cleaner production. | Classical TPM | [14] |
| | TPM is a lean manufacturing tool that supports the industry's economic sustainability. | Classical TPM | [86] |
| | TPM is among the lean–green and sustainability (LGS) tools that aim to achieve a superior triple bottom line (TBL) and positively impact the company's economic and environmental pillars. | Classical TPM | [87] |
| | TPM is a methodology to improve the availability, productivity, and quality of manufacturing systems. | Classical TPM | [39] |
| | TPM is a methodology to maximize equipment effectiveness by actively involving all supporting departments. | Classical TPM | [40] |
| | TPM is a methodology aiming to increase equipment and machinery efficiency and longevity. | Classical TPM | [93] |
| | TPM is the most effective maintenance strategy to improve equipment availability and product quality while reducing waste. | Classical TPM | [34] |
| | TPM is a strategy to reduce equipment failure, minimize solid waste generation, and boost machine efficiency. | Classical TPM | [38] |
| | TPM is a maintenance philosophy that helps businesses improve their operational performance by acting on diverse dimensions such as productivity, quality, safety, flexibility, and costs. | Classical TPM | [43] |
| | TPM is a method that can be used to control operational performance and bring out improvements in production in the era of Industry 4.0. | Classical TPM | [95] |
| | TPM is a robust maintenance management approach grounded on lean principles. | Classical TPM | [11] |
| | TPM is a strategic management initiative that improves the machine lifecycle and productivity. | Classical TPM | [33] |
| | TPM is a high-effectiveness approach to maximize production in any industry. | Classical TPM | [94] |
| | TPM is a productivity improvement program for various manufacturing industries. | Classical TPM | [44] |
| | TPM is a system that maintains and improves business production and quality systems in terms of integrity by acting on equipment, processes, and people rather than making new investments. | Classical TPM | [36] |
| | TPM is an intellectual project that goes beyond the methodology or strategy of a firm to act on its ideology and anchor the manufacturing system sustainability mindset throughout the organization as a whole (concept of STPM). | STPM | [18] |
| 2020 | TPM is a lean tool that improves business productivity by reducing waste to meet customer demand, which justifies its significant positive impact on green manufacturing. | Classical TPM | [17] |
| | TPM is a lean tool that helps businesses reach economic and environmental gains and improve environmental management, which allows them to stand out from the competition and boost revenues. | Classical TPM | [90] |
| | TPM is a methodology to banish losses due to inefficiencies. | Classical TPM | [45] |
| | TPM is a methodology that prioritizes eliminating efficiency losses and uses some maintenance activities from preventive maintenance. | Classical TPM | [46] |
| | TPM is a methodology that aims to improve maintenance management and ensure the best operational performance for assets. | Classical TPM | [146] |

**Table 2.** *Cont.*

| Year | Definition of TPM/STPM | STPM/Classical TPM | Source |
|---|---|---|---|
| 2020 | TPM is a cutting-edge maintenance strategy that will provide a comprehensive understanding of strategic maintenance. It can be scaled to Green TPM as an integrated approach covering elements like green training, maintenance, and six sigma, supporting higher manufacturing and environmental performance. | Green TPM and transition toward STPM | [16] |
| | TPM can be characterized as a manufacturing strategy to raise product quality and equipment productivity. | Classical TPM | [53] |
| | TPM is a systemic approach that emphasizes improving the efficiency of the manufacturing system; it constitutes one of the cornerstones of business management culture. | Classical TPM | [111] |
| | TPM is a vital tool for improving manufacturing firms' productivity. | Classical TPM | [47] |
| | TPM is a modern maintenance practice that supports industrial production systems by reducing breakdowns, defects, accidents, and waste. | Classical TPM | [48] |
| | TPM is a technical practice; its adoption can be positively influenced by two support factors, top management involvement/leadership and human resource development within manufacturing firms. | Classical TPM | [50] |
| | TPM is among the performance improvement techniques widely deployed within manufacturing businesses and aims to achieve a competitive advantage, economic viability, customer fulfillment, dependability, and survival. | Classical TPM | [52] |
| | TPM is among the best waste management techniques. | Classical TPM | [89] |
| | TPM is a continuous improvement program that allows firms to enhance their performance and competitive advantage while achieving environmental sustainability. | Classical TPM | [88] |

Table 2 shows that until the last three years, most researchers only knew of TPM and defined it in different ways as classical or Japanese TPM based on the eight pillars (Section 3.4), thus lacking the concept of sustainability. Based on this table, in the period of 2020–2022, 94% of the selected papers studied classical TPM, and only 4% emphasized the new concept of STPM, which were carried out by the authors of [18,19] in 2021 and 2022; the remaining 2% is represented by [16], which led the transition from TPM to STPM in 2020 through the creation of Green TPM as a transitional concept toward STPM. Furthermore, until today, Morocco and the USA were the pioneering countries, with one paper each emphasizing the concept of STPM in the industrial literature.

*3.4. Mechanisms of STPM*

The eight pillars of TPM are focused improvement, autonomous maintenance, planned maintenance, training and education, early equipment maintenance, quality maintenance, TPM in administration, and safety, health, and environment [19]. They are dedicated to maximizing the effective production of any industry [94]. TPM shares only four pillars with STPM; the latter's fifth and sixth pillars are all one team and sustainability maintenance, respectively (Figure 7).

Starting with sustainability maintenance (i.e., the sixth pillar of STPM) as a sustainability pillar, it aims at improving the triple bottom line of sustainability by acting on the maintenance function in the context of R4thGM because it seems impossible to introduce a sustainable philosophy (e.g., STPM) into an organization without applying a sustainability-driven management paradigm. Sustainability maintenance targets the lack of sustainability in TPM (Section 3.3). This new pillar includes and goes beyond the two pillars "training and education" and "safety, health, and environment," for classical TPM to encompass sustainability's economic, social, and environmental foundations (Table 3). It allows for sustainable maintenance and fosters the company's sustainability due to its alignment with the business orientation toward sustainability and customers (Figure 4). Sustainable maintenance, as one of the core aims of this pillar, should contribute to minimizing the

environmental and social impacts of a manufacturing system, reducing life cycle costs, enhancing durability for equipment, and improving socio-economic well-being [7,147]. Table 3 is supported by [6,16,148–150].

**Table 3.** STPM's sustainability maintenance pillar and its role in improving the triple bottom line of sustainability for manufacturing companies.

| Sustainability Bottom Line | Improvement Measure | Role of Sustainability Maintenance (Sixth Pillar of STPM) in Improving Sustainability |
|---|---|---|
| Economic | Optimizing maintenance operation costs and manufacturing and resource consumption costs by acting on the maintenance function. | Selecting the optimal maintenance strategy. |
| | | Encouraging and sustaining accessible, affordable, dependable, cleaner, and modern energy systems for the company. |
| | | Setting and optimizing economic indicators for the maintenance system [149]:<br>- Costs related to sustaining environment, health, and safety (EHS) compliance, including penalties, liabilities, worker compensation, the cost of control equipment and its depreciation, remediation costs, and personnel costs.<br>- Prices of energy consumed by maintenance processes (tools, means of transport, etc.).<br>- Costs for maintenance waste treatment.<br>- Costs of storing and recycling spare parts.<br>- Prices for the purchase of maintenance tools.<br>- Fees for the acquisition of spare parts, materials, and consumables.<br>- Costs of maintenance employees.<br>- The ratio of actual labor hours to planned labor hours in a maintenance operation.<br>- Costs of investments to develop the maintenance function (R&D, maintenance infrastructure, etc.). |
| | Supporting the circular economy from a maintenance perspective. | Closing the loop and continuous improvement of the circularity of materials and energy (spare parts, consumables, water, electricity, heat, gas, etc.) deployed in the maintenance of the manufacturing system. |
| | | Building and broadening a symbiosis network to facilitate free-of-charge exchanges of manufacturing and maintenance releases necessary (e.g., as raw materials) for production. |
| | | Fostering circularity-driven decision making by considering all internal and external stakeholders (including the maintenance employees) in the open innovation process. |
| Social | Providing decent work for all employees and managing the social environments (i.e., diverse interacting stakeholders, suppliers, employees, shareholders, etc.) of the maintenance system. | Ensuring employees' well-being within and beyond the company. |
| | | Setting and optimizing social indicators for the maintenance system [149]:<br>- Health and safety of maintenance employees, including the type and rate of injury, occupational disease rate, absence rate, missed workday rate from maintenance-related illnesses or accidents, and work-related fatalities from maintenance-related injuries or occupational diseases.<br>- Programs for helping maintenance workforce employees and their families with serious diseases include education, training, counseling, risk control, and prevention measures. |
| | Improving the working climate. | Motivation. |
| | | Developing communication mechanisms and eliminating barriers at all company hierarchy levels and beyond it (i.e., between it and its external stakeholders). |
| | | Providing training and education for all employees. |
| | | Believing in human values and developing them. |
| | | Creating an open, inclusive, and equitable environment for employees. |
| | | Encouraging creativity and open innovation while considering all company's stakeholders (including maintenance employees) as essential decision makers to achieve cross-functional operational, tactical, and strategic goals. |

**Table 3.** *Cont.*

| Sustainability Bottom Line | Improvement Measure | Role of Sustainability Maintenance (Sixth Pillar of STPM) in Improving Sustainability |
|---|---|---|
| | Developing the societal values of corporate citizenship. | Making the company a social sub-system at the heart of societal activities and events and active within its changing international environments in the framework of its social–civil accountabilities. |
| Environmental | Sustaining the continuous improvement of environmental performance for the company as an ecosystem by acting on the maintenance function. | Setting and optimizing the different families of environmental indicators for the maintenance system [149]:<br><br>- Material-related environmental indicators.<br>- Waste-related environmental indicators.<br>- Emissions-related environmental indicators.<br>- Energy resource-related environmental indicators. |
| | | Encouraging clean energy sources in maintenance activities. |
| | | Supporting the transition from fossil fuels to green resource consumption for maintenance. |
| | | Fostering environmental sustainability-driven decision making by involving all company stakeholders, including maintenance employees, in the open innovation process. |
| | | Increasing employees' understanding of climate change and all of the company's external context-related environmental issues. |
| | | Supporting the circular economy paradigm from a maintenance viewpoint through the following:<br><br>- Closing the loop of materials and energy (spare parts, consumables, water, electricity, heat, compressed air, gas, etc.) deployed in maintenance.<br>- Creating an industrial symbiosis network to ensure the exchange of maintenance releases.<br>- Enhancing green maintenance by applying green techniques [14] such as reducing materials and energy consumption for maintenance, recycling and using friendly recyclable materials in maintenance, recovering maintenance resources by straightening their longevity, etc.<br>- Broadening the circular economy dimensions, which are regenerate, share, optimize, loop, virtualize, and exchange (i.e., the ReSOLVE framework) [150]: regenerate—shift to renewable materials and energy to use in maintenance; share—reuse spare parts throughout their technical lifetime (i.e., second-hand) and prolong their life through maintenance, repair, and design for durability in general and environmental performance in particular; optimize—remove waste due to the abnormal functioning of machines or low performance in any phase of the supply chain of materials to consider for maintenance; loop—keep maintenance materials, components, and energy in a closed loop (recycling them, keeping them renewable, etc.); virtualize—use e-maintenance, online maintenance services, predictive maintenance 4.0, digital twin/artificial intelligence/blockchain/big data analytics-based maintenance (i.e., maintenance 4.0 in general), zero paper, etc., to reduce carbon footprint and enhance general environmental performance; exchange—replace old materials or parts dedicated for maintenance with advanced non-renewable ones, use new technologies, etc. |
| | | Taking actions for green training, maintenance, equipment, and shop floor orientation to develop a green culture likely to help the company significantly achieve environment-related cross-functional goals; this framework was termed Green TPM [16]. |

All one team (i.e., the fifth pillar of STPM) is a corner of Joiner's triangle [5] that has recently been explained through Hallioui's triangle (Figure 4) as "engagement of people, including leadership" and "relationship management" [2]. In the scope of STPM (Figure 7), the engagement of people, including leadership, means the total involvement of all internal stakeholders of the company (i.e., all employees, including managers and non-managers, from all organizational classes and departments, shareholders, etc.) in achieving the goals of STPM. Relationship management is the dedication to managing relationships with internal and external stakeholders of the company to achieve and sustain STPM. This STPM pillar replaces the TPM in administration pillar from classical TPM since it considers administration and other departments independent from each other

(i.e., without interactions). In other words, it proves bureaucracy has had a place among traditional companies with no openness to diverse stakeholders or sustainability awareness among them [2].

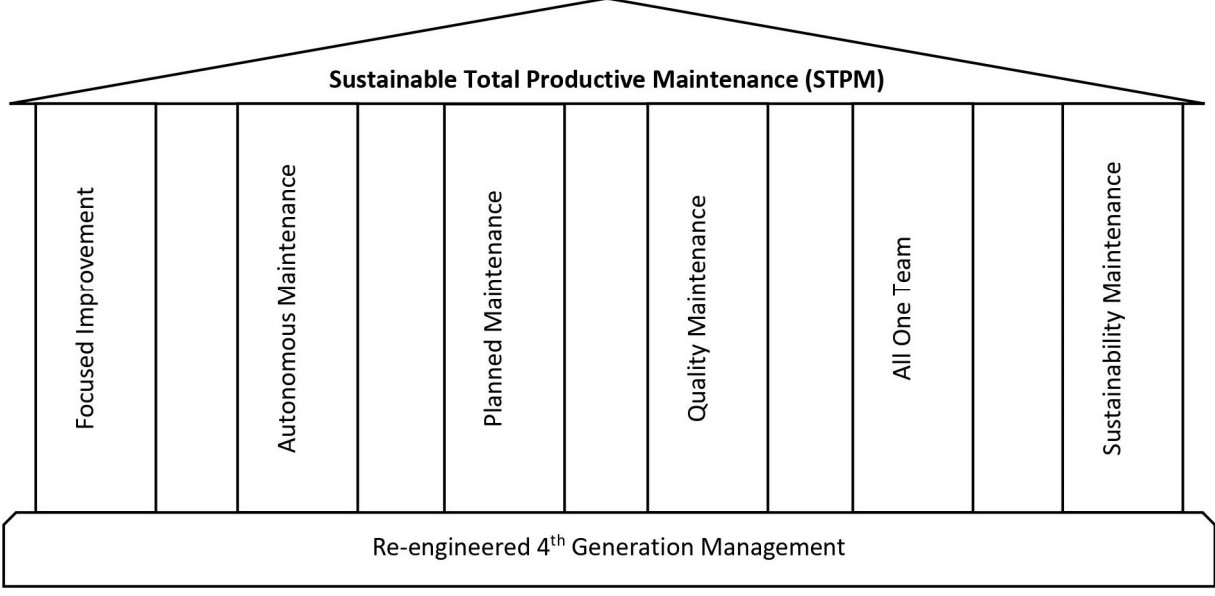

**Figure 7.** The suggested unique architecture of Sustainable Total Productive Maintenance.

The four remaining pillars are well known in the lean manufacturing literature and common between TPM and STPM (Figure 7). This includes focused improvement, which relies on using problem-solving methods to encourage employee participation and suggestions (e.g., Kaizen) to ensure continuous improvement in equipment performance, thereby continuously reducing waste [19,23]. Moreover, this pillar aims to assess the operational effectiveness of equipment based on Overall Equipment Effectiveness (OEE). Autonomous maintenance is dedicated to including small maintenance tasks (i.e., first and second maintenance levels) into the duties of operators to make them more familiar with machines, encourage them to be more involved in the operational performance of their line, and improve their consciousness in terms of machine effectiveness and its impact on quality production, which enhances the focus and efficiency of maintenance personnel. Planned maintenance includes preventive maintenance (i.e., time-based maintenance) and even predictive maintenance as condition-based maintenance [79] to achieve zero failure. Quality maintenance aims at maintaining and continuously improving the quality of the products by mastering the machines, methods, materials, workforce, and environment; it requires the application of the tools necessary for continuous quality improvement (5M, statistical process control tools, etc.) to achieve zero quality defects.

## 4. Propositions and Research Implications

### 4.1. Propositions to Achieve STPM

#### 4.1.1. Applying R4thGM First

The sort, set in order, shine, standardize, and sustain methodology (5S) as a lean tool to enable a waste-free-driven organization for manufacturing processes [151] has long been a support technique and prerequisite to implementing classical TPM. Operationally, we cannot disregard its essential role at the preliminary stage of implementing STPM. However, R4thGM is the primary condition, the most adequate management style, and the most favorable environment for deploying any sustainability philosophy or systems approach, such as STPM (Figure 7), in the second era of contemporary companies [2]. Hallioui's triangle supports the six pillars of STPM because sustainability and customer focus, systemic and evidence-based decision making, engagement of people including

leadership and relationship management, and continuous improvement for sustainable business performance and evolutionary goals in a changing external competitive environment as foundations for it are all in alignment with sustainability maintenance, all one team, quality maintenance, planned maintenance, autonomous maintenance, and focused improvement, which are the six pillars of STPM (see Figure 7 and Section 3.4).

4.1.2. Using Industry 4.0 Technologies to Enhance the Pillar of Sustainability Maintenance

Since R4thGM creates and maintains the sustainability orientation for companies, this subsection focuses only on supporting sustainable maintenance through Industry 4.0 technologies. Recent studies have shown an increasing interest in maintenance and sustainability and the significant potential of Industry 4.0 digitalization technologies as key enablers of sustainable maintenance [7]. The maintenance function has begun to transform its role to better enhance value creation by contributing to modern manufacturing companies' economic, environmental, and social dimensions of sustainability through maintenance 4.0 [100]. Maintenance 4.0 is one of the features of contemporary maintenance.

There is a scarcity of literature studying the new concept of contemporary maintenance, but maintenance 4.0, or smart maintenance, has recently become well known among academics and practitioners. It is often defined as Industry 4.0-based maintenance and includes a range of methods for monitoring the current condition of machines to predict upcoming machine failures through the use of automated real-time analytics and supervised or unsupervised machine learning to prescribe an optimal course of action in real time, thus analyzing potential decisions and their interactions [98,152], which can be summarized as the reduced frequency of maintenance and processing data in real time to deliver pertinent information [153]. In the framework of the pillar of sustainability maintenance (Table 3), maintenance 4.0 as a strategy of contemporary maintenance can enable contemporary companies to extend the useful life of their manufacturing equipment while minimizing planned downtime, preventing unplanned downtime, improving process and worker safety, and consuming as little energy and resources as possible while also reducing costs [100]. The use of digital technologies in sustainability management, in general, is still very limited and is currently mainly in the pilot phase [154]. Here are some Industry 4.0 technology-enabled solutions (e.g., digital twins and artificial intelligence) in the recent literature aimed at the pillar of sustainability maintenance that enhance sustainable maintenance and, therefore, the focus on sustainability for manufacturing companies (Figure 7 and Table 3):

Digital twins (DTs) allow for the creation of a dynamic digital replica of a physical system [99,155]. This is one of the main tools supporting sustainable maintenance and manufacturing activities [96]. This digitalization technology improves the system's connectivity and flexibility, besides increasing its intelligence due to the introduction of knowledge and experience to a computer program; it provides the possibility of conducting simulations to amend manufacturing process irregularities and account for sustainable maintenance [99]. Evidence from empirical studies on digital twin-enabled sustainable maintenance in the manufacturing industry is still scarce in the literature. To improve companies' sustainability, the authors of [96] suggest using digital twin models in maintenance and manufacturing activities. In addition, the authors found that most studies in the literature tackle only two dimensions of sustainability, i.e., the economic and environmental dimensions, which is not the case for our study (Table 3).

Artificial intelligence comprises machine learning and natural language processing technologies, enabling machines to feel, understand, act, and learn [2,156]. So far, studies on using it as a tool for sustainable maintenance within manufacturing businesses are still scarce. In the intersection between this Industry 4.0 technology and sustainable maintenance in the context of the manufacturing sector, predictive maintenance has become a potential maintenance strategy that can benefit manufacturing through better sustainable solutions due to the incorporation of the Industrial Internet of Things (IIoT) [157]. To present a new level of agility to cyber-assisted maintenance activities and full lifecycle

consideration of assets while emphasizing the necessity of achieving sustainability goals for manufacturing companies worldwide, the authors of [97] highlighted contemporary maintenance from an ergonomic viewpoint by bridging the gap between humans and machines through Explainable Artificial Intelligence (XAI) that provides artificial intelligence algorithms with narrative functionality so that they can communicate the significant steps taken in solving the problem to a human. Indeed, they presented a framework for human-in-the-loop intelligent and sustainable maintenance, which is, so far, one of the most important samples of contemporary maintenance in the manufacturing literature; in this artificial intelligence-enabled sustainable maintenance solution, XAI was used since it refers to developing artificial intelligence models that are transparent and understandable to humans. Operationally, XAI focuses on designing artificial intelligence systems that can provide clear and interpretable explanations of their decision-making processes rather than relying simply on the opaque black-box models that are frequently used.

### 4.2. Results and Discussion

Since the authors of [18]'s proposal of creating STPM, the only existing research work studying it was carried out by [19] (Table 1), which defines it as a complement to TPM that could serve as a vital tool for more sustainable manufacturing, in other words, STPM is presented as being founded on the eight pillars of TPM, which is not in alignment with the perspective of the STPM idea engineered in 2021 (Figure 3). Besides showing that the root causes of failure to implement TPM across manufacturing companies are managerial barriers (Figure 6), and to prove the cruciality of the managerial framework (i.e., management style) in implementing industrial systems approaches such as TPM and that its eight pillars are insufficient to overcome those barriers (Section 3.2), we confirmed the lack of sustainability in the concept of TPM (Section 3.3) and the scarcity of literature discussing STPM (Tables 1 and 2). The main objective of this study was to present STPM as a substitute for TPM while suggesting its unique formalism based on R4thGM (Figure 7). The overall study process (Figure 3) was designed to support the sustainability orientation of manufacturing companies from a sustainable maintenance viewpoint in the second era of contemporary organizations (Section 1); STPM is a crucial enabler of sustainable maintenance and encourages contemporary maintenance and contemporary businesses. From the Scopus, Web of Science, and Science Direct databases, we chose 94 relevant papers published between 2008 and 28 February 2023 (Table 1). We found that 77% of relevant papers were published during the period of 2018–2023. Three relevant journal papers were published in this period in Sustainability (Switzerland), ranking it second behind the Journal of Quality in Maintenance Engineering, which published four relevant papers. However, considering the research topic of this article, Sustainability (Switzerland) is regarded as the most pertinent journal to this study.

Through this work, concepts found in the lean–green manufacturing literature are gathered under the umbrella of STPM, such as Green TPM, green manufacturing approaches (6R techniques, the ReSOLVE framework, circular economy dimensions, etc.), and green maintenance supporting the environmental performance of companies. The latter goes beyond enhancing the environmental foundation of sustainability to strengthen corporate sustainability's economic and social bottom lines, which is based mainly on the novel sustainability maintenance pillar (Figure 7 and Table 3). This STPM pillar is created to achieve sustainable maintenance and foster the manufacturing businesses' sustainability orientation, which is catalyzed by R4thGM as a managerial catalyst for the six pillars of STPM (Section 3.4) and the necessary conditions to implement STPM (Section 4.1.1). Furthermore, based on recent industrial studies, we suggest using Industry 4.0 digitalization technologies to enhance STPM's sustainability maintenance pillar, and research outcomes show that digital twin and artificial intelligence-based solutions for sustainable maintenance are rare and the only existing solutions in the literature (Section 4.1.2). In other words, only scarce digital twin and artificial intelligence-driven contemporary maintenance solutions are found.

Besides the sustainability maintenance pillar, all one team is also a new pillar built for STPM (Figure 7 and Section 3.4) to broaden the total involvement impact (including top management commitment influence) and eliminate barriers (i.e., strengthen interactions) between departments (including administration) characterizing traditional organizations and their bureaucracy (Section 3.2). All one team is suggested as a substitute for the TPM pillar called TPM in administration, which can be regarded as a major promoter of the lack of top management commitment and support, which represents a significant barrier to implementing TPM (Section 3.2). In addition, it is a new foundation for increasing understanding of the systems approach and developing system spirit (i.e., all working within a system) among manufacturing organizations. STPM and TPM have only four common pillars—focused improvement, autonomous maintenance, planned maintenance, and quality maintenance (Section 3.4).

From this work onward, STPM will be studied and implemented as a substitute for TPM within manufacturing businesses, as intended by the authors of [18]. Academics and practitioners can perceive the substantial difference between STPM and TPM in form, finality, and origin (Sections 3.1 and 3.4, Figure 5). STPM supports the triple bottom line of sustainability for companies (Table 3) while encompassing improvements in operational performance dimensions such as productivity, quality, and costs, which is reachable through TPM according to many studies founds in the literature, as shown in Table 2.

Moreover, this work is the pioneer in presenting STPM as a substitute for Japanese or classical TPM, proposing and describing the unique architecture of STPM, and suggesting the concept of STPM as a compromise between sustainable maintenance, sustainability maintenance, contemporary maintenance, R4thGM, and second-era contemporary manufacturing businesses, which have never been gathered in the lean manufacturing and sustainability literature. Thus, for manufacturing managers and policymakers, it will serve as the primary guide to end the age of TPM and its implementation barriers by replacing it with STPM as an enabler of sustainable maintenance and a catalyst for Industry 4.0 technology-based (digital twins, artificial intelligence, etc.) sustainable maintenance (i.e., contemporary maintenance). Therefore, this represents an adequate philosophy or industrial systems approach to second-era contemporary businesses. Still, STPM is a new lean manufacturing and sustainability key that will support the transition launched in 2022 from first-era manufacturing organizations toward second-era ones. It will help policymakers and managers set the ideology of sustainability and its foundation within the circular economy from a maintenance perspective within companies (Table 3).

Furthermore, significant support from top management in achieving STPM through (i) the application of R4thGM, known for its substantial implications on considering and advancing the sustainable triple bottom line and realizing the 17 Sustainable Development Goals (SDGs) defined in the UN 2030 Agenda for Sustainable Development [2], and (ii) the use of Industry 4.0 technologies to enhance the STPM sustainability maintenance pillar will be beneficial to manufacturing companies in terms of supporting their countries to achieve SDGs such as SDG 3 (ensure healthy lives and promote well-being for all at all ages), SDG 6 (ensure availability and sustainable management of water and sanitation for all), SDG 7 (ensure access to affordable, reliable, sustainable, and modern energy for all), SDG 8 (promote sustained, inclusive, and sustainable economic growth, full and productive employment, and decent work for all), SDG 9 (build resilient infrastructure, promote inclusive and sustainable industrialization, and foster innovation), and SDG 12 (ensure sustainable consumption and production patterns).

## 5. Conclusions and Future Research

This paper reviewed 94 papers to present STPM as a novel substitute for TPM while upgrading the latter to current economic, environmental, and social trends by proposing the unique six-pillar architecture of STPM based on R4thGM. It addresses the scarcity of literature on STPM. In addition, it aligns the understanding of STPM with the perspective of its idea built in 2021 while dealing with the challenge of making STPM a compromise

between modern concepts, such as sustainable maintenance, contemporary maintenance, R4thGM, and second-era contemporary manufacturing businesses. STPM is a catalyst for Industry 4.0-based sustainable maintenance and goes beyond enhancing sustainability's environmental foundation to strengthen corporate sustainability's economic and social bottom lines, which will support the first–second era transition for contemporary manufacturing firms and fortify their involvement in achieving SDGs. Implementing R4thGM first and then using Industry 4.0 technologies was proposed to achieve STPM. Furthermore, using digital twins and artificial intelligence as tools for sustainable maintenance in the manufacturing sector remains at the proposal level for future research contributions.

The three-stage qualitative research process used to perform this critical review article (Figure 3) allowed us to explore factual limitations attributed to a lack of secondary data in the literature, which were justified by the novelty of the research topic and its components. So far, no studies or experimental evidence (e.g., case study findings) have been made available regarding the implementation of R4thGM, and—second-era contemporary companies launched in 2022 have still not been investigated. There was also no experimental feedback to help us provide more propositions to achieving STPM while dealing with its unknown implementation barriers. There is also a scarcity of studies discussing the concept of contemporary maintenance, and there remains a lack of research into studying the role of maintenance in improving the circular economy or the relationship between them; there is only one study carried out by [158]. There was no evidence from empirical studies exploring blockchain technology, cloud computing, or big data analytics-enabled solutions in relation to enhancing sustainable maintenance in the manufacturing sector (i.e., a lack of blockchain-driven, cloud computing-driven, or big data analytics-driven contemporary maintenance solutions in recent studies). The literature does, however, provide rare and straightforward discussions on the technology aspect of digital twin- and artificial intelligence-based solutions to support the STPM sustainability maintenance pillar (Figure 7) and therefore strengthen sustainable maintenance and even contemporary maintenance.

Refs. [18,19] are the two papers discussing STPM in the literature (Table 1). Further studies can be carried out to assess the effectiveness of STPM compared with TPM to improve sustainable performance for manufacturing companies and lead their orientation toward sustainability by fostering sustainable maintenance and encouraging contemporary maintenance as a lever of second-era contemporary organizations. In addition, empirical studies could help us locate resistance to change in deploying STPM within manufacturing firms; in principle, barriers to implementing STPM would serve as an outline, impacting businesses' focus on sustainability and inhibiting the support of countries in achieving SDGs. However, at first, more work is needed to study contemporary maintenance and emphasize the capabilities of Industry 4.0 technologies such as digital twins, artificial intelligence, blockchains, cloud computing, and big data analytics when incorporated into the circular economy to lead the transition of companies toward the second era of contemporary businesses while investigating the centric role of R4thGM. Moreover, other future studies might help prove the importance of re-engineering lean manufacturing strategies (i.e., creating novel lean manufacturing and sustainability strategies such as STPM) and basing them on R4thGM to respect and develop sustainability's triple bottom line in the manufacturing sector.

Ultimately, STPM seeks to increase the probability of achieving and maintaining higher levels of sustainable and operational performance by identifying, prioritizing, and eliminating the causes of the losses. We suggest that the causes cannot be fully understood without understanding the systems' ideas [159]. The foremost of these ideas is viability [160] in the context of general systems theory. However, the literature has not yet discussed the relationship between STPM and viability. Research emphasizing these two concepts would open the eyes of academics and encourage manufacturing businesses and their stakeholders to reach an excellent triple bottom line of sustainability. To optimize the equipment lifecycle, artificial intelligence, digital twins, or artificial intelligence-driven digital twins might cover

the complete equipment lifecycle, from design and production to operation and end-of-life recycling. Therefore, manufacturing firms might mitigate environmental impacts and promote circular economy dimensions by considering sustainability factors across the lifespan. Investigations on STPM as a form of leverage for the circular economy could be conducted to help 21st-century manufacturing ecosystems build and broaden their industrial symbiosis networks and support the survival of organizations in a VUCA world.

Moreover, the current North American and European manufacturing sectors are threatened by the increasingly stifling aging workforce phenomenon; based on Bureau of Labor Statistics data for 2022, the median age of manufacturing workers in the United States is 44.3 years [161], while today, one in five Europeans is 65 years or older, which will be close to 30% by 2050 [162]. In the framework of investigating STPM in general or sustainability maintenance for organizations in the era of Industry 5.0, studies on augmented reality, virtual reality, and other human augmentation tool-based sustainable maintenance could play a pivotal role in supporting the efforts of organizations and maintenance managers toward coping with the aging phenomenon in the future that will require massive upskilling and human-centered support technology. Surveys should be conducted to assess the impact of STPM on sustaining the aged workers in the manufacturing sector while studying its role in supporting companies in developing an aging maintenance workforce that is productive, healthy, and capable.

**Author Contributions:** Conceptualization, A.H., B.H., P.F.K., R.S.S., O.E. and M.J.-K.; methodology, A.H.; software, A.H.; validation, M.J.-K., P.F.K., O.E., B.H., R.S.S., P.C.M. and J.M.S.; formal analysis, A.H.; investigation, A.H.; resources, A.H.; data curation, A.H.; writing—original draft preparation, A.H.; writing—review and editing, A.H., O.E., M.J.-K., P.F.K., R.S.S., J.M.S. and P.C.M.; visualization, A.H. and B.H.; supervision, A.H., B.H., P.F.K., R.S.S., O.E. and M.J.-K.; project administration, A.H. and B.H. All authors have read and agreed to the published version of the manuscript.

**Funding:** This study received no specific financing from public, corporate, or non-profit funding agencies.

**Institutional Review Board Statement:** Not applicable.

**Informed Consent Statement:** Not applicable.

**Data Availability Statement:** The data presented in this study are available on request from the corresponding author.

**Acknowledgments:** The authors sincerely appreciate the MDPI reviewers' time and insightful comments.

**Conflicts of Interest:** The authors state that they are aware of no personal or financial conflict that might have affected the research reported in this study.

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
