# Peer review of "A Review of Sustainable Total Productive Maintenance (STPM)"

_sustainability, doi:10.3390/su151612362_

Round 1

Reviewer 1 Report

The topic of this paper is very interesting and demanding, particularly in the industry 4.0 era. The authors have done a great job compiling sufficient data on total productive maintenance. This paper highlights the importance of sustainability for maintenance as well and introduced a new term STPM. There are some minor issues in this paper that are given below and I recommend to address to improve the overall quality of this manuscript. 

Major issues:

1. The length of the paper is too long due to some poor management/arrangement of tables and figures. I think the issue can be resolved like Figure 8 and Figure 9 is taking more space and providing comparatively less information and that information is even described in the previous table as well. The author should review the whole paper keeping in view this problem. The manuscript can be improved by removing the dual explanations.   similarly, Table 3 should be revisited to improve the presentation. 

2. The outcome of any research paper is new research directions for its readers. But i am surprised that in this paper author did not presented the conclusion section separtely and there is not new or emerging research directions for the researcher of this area. I strongly recommend the authors to add the conclusion section and a subsection if emerging research areas and explain them in detail.  

Minor issues:

1. Caption of Figure 1, make it "Number" not "Namber".

2. Why is the caption of Figure 1 and figure 2 the same.?

3. There are grammatical errors throughout the document. Revisit the manuscript for this purpose also. 

Need to review the document to remove grammatical errors. 

Reviewer 2 Report

Figure 5 needs a little revision in order to better understand the connection with figures 1 and 2. It should be mentioned that, in 2015 and 2016, the  research results are starting to be published ")

254 - I think the introduction of the TQM concept should be preceded by a brief presentation of the emergence of Quality Management Systems (one or two sentences)

319 - I think it is difficult to follow the reference to figure 10, very far from the text (I think this reference can be abandoned)

472 - I believe that the 5S concept (detailed name) should be specified for familiarizing the non-specialist reader

The work, as a whole, is interesting in terms of analyzing the specialized literature and supporting the idea of STPM.

The concept of using artificial intelligence as a tool for sustainable maintenance requires more support with bibliographic references or personal experience. Otherwise, I think it can only remain at the level of proposal for future studies.

A personal opinion, which does not have the role of detracting from the value of the work or contradicting the authors: I think that the statement "STPM is a substitute for TPM" is a bit harsh. I believe that STPM is an upgrade (updating) of TPM to the current economic and social trends, respectively sustainable TPM

Reviewer 3 Report

Line 25,26,27:

What are the characteristic of industries that have implemented STPM?.

Line 167:

Search item by: STPM, only consist of 2 relevan papers, so these articles need to be discussed in great detail in designing the future research framework.

Line 399 (Table 2): It would be more practical if the definition of TPM were grouped (for example: TPM as a strategy and TPM as a tool, or something else).

Need editing of English language.

Reviewer 4 Report

The authors present a literature review of Sustainable Total Productive Maintenance.

1.       Why don’t you state the exact number of papers found instead of >1000?

2.       What filtering methodologies did you use (figure 3). You don’t mention these in the text.

3.       Please state how many papers have been identified by each search engine.

4.       I am not sure whether the key words were selected appropriately: "Lean methods and sustainability" will only return papers where the abstract or title stated the exact phrase. Scopus returns no papers for this search term. When selecting key words, I would recommend checking whether each of the key word used does return results when used individually.

5.       Figure 3, you state that you used google scholar to identify peer-reviewed publications. However, in the methodology section, you did not mention that you used google scholar.

6.       The key words you mention in the text are not the same key words you mention in figure 3. And the key words you mention in table 1 are also different from the other two sets of keywords. Which key words did you use?

7.       Line 157, is the semicolon supposed to be a comma or a full stop? If it is a full stop, why do you specifically select papers that discuss “contemporary maintenance” but don’t include “contemporary maintenance” as a keyword?

8.       Have the papers been categorised into relevant and not relevant by multiple authors? What was the interrater reliability?

9.       Table 1, please state the number of papers identified in each of the databases and how many duplicates were there.

10.   Based on table 1, no papers have been identified by two different searches. This is highly unlikely. You have the search term “TPM” linked with an “or” in the first search and in the second search you have “(TPM & Sustainability)” linked with an “or” with other search terms. Any paper identified by the (TPM & Sustainability) search term, should also be identified if TPM is used by itself as in the first search.

11.   Figure 9, was there only two apers published? The map only shows two countries/areas which each have the number 1 written in them.

Round 2

Reviewer 1 Report

All the suggestions and recommendations have been addressed by the authors. Thus, the reviseed manuscript can be accepted. 

Author Response

We sincerely appreciate your time and insightful recommendations. Thank you, Reviewer 1, for helping us improve our paper.

Reviewer 4 Report

The authors have addressed all my comments.

However, I still find it strange that there are no duplicates between the categories (i.e. none of the papers identified through the search terms in category 1 have been identified in any of the following 5 categories). I doubt that: Category 2 includes the search term “tpm AND sustainable AND manufacturing”. Any papers identified through this search term should also be identified with the category 1 search term “tpm AND sustainability”. However, you state that this would not be the case.

Have you really reviewed the abstract of more than 31846 papers to categorise them into relevant or irrelevant? Or did you use a computer programme to do this?
